# Establishing Picture Databases for Image Boards: An Example for Lifestyles of Health and Sustainability Images

**Peng-Jyun Liu [1,2,*], Ming-Chuen Chuang [1] and Chun-Cheng Hsu [1]**

[1] Institute of Applied Arts, National Chiao Tung University, Hsinchu 300, Taiwan; cming@faculty.nctu.edu.tw (M.-C.C.); huncheng@nctu.edu.tw (C.-C.H.)

[2] Department of Creative Product Design, Asia University, Taichung 413, Taiwan

[*] Correspondence: a804338@gmail.com

**Abstract:** Recently, more importance has been attached to consumers' emotional feelings in the course of product design. Designers must convey positive emotions, such as surprise and affection, to consumers through their designs. For this purpose, image boards have been frequently used in design to position product emotional feeling and arouse design ideas. A large number of pictures are often needed for constructing an image board. However, it is time-consuming and labor-intensive to find appropriate pictures and the pictures that are finally collected may not reflect the expected image of consumers. Therefore, this study aims to take Lifestyles of Health and Sustainability (LOHAS) as an example to build a user-driven database for image boards. In this research, 16 LOHAS representatives were identified and recruited by using a lifestyle questionnaire to collect, and then screen out 50 proposed pictures relevant to the image of LOHAS. Since image boards are usually used by designers, in order to include their ideas, another 16 pictures were selected by the invited experienced product designers to create a comprehensive pool of 66 proposed pictures. Design experts were asked to select six key image adjectives, which included healthy, environmentally friendly, sustainable, natural, simple, and ecological for describing images of the LOHAS, from the vocabulary pool collected by general respondents, LOHAS representatives, and designers. Next, 219 LOHAS subjects were required to carry out a semantic differential assessment of each of the 66 proposed pictures with the six key images, and then two types of analyses on the collected data from the semantic differential assessment. Through mean analysis and grey correlation analysis, the recommended pictures representing LOHAS or six key adjective images were selected. The research results put forward three database application models. The results of this study are expected to be used by designers, users, manufacturers, and educators to help improve product design efficiency in the future.

**Keywords:** picture database; image board; semantic differential scale; grey relational analysis; product design; user experience

## 1. Introduction

One of the goals that designers strive for is to integrate positive emotion into product design, because it can stimulate consumers' motivation to purchase. Roy, Goatman, and Khangura [1] pointed out that positive and joyous emotions are one of the key points for product design. Lauer and Pentak [2] claimed that in the design process, implementing the proper method or tool helped create a successful design. Green and Bonllo [3] also stressed that a proper design process and method helped define ideas and make decisions. Design methods help to stimulate design thinking, obtain diversified design inspirations, and assist designers' understanding of the available design elements. Among available

design methods or tools, one of the commonly used tools that designers use to realize and satisfy design is an image board.

An image board was first proposed, in 1995, by Baxter, who pointed out that he could investigate the empathy effect by making an image board to fully understand users' feelings [4]. An image board is a type of design tool that uses intuitional judgment to lets designers directly put themselves in the shoes of users to experience design problems and avoid design fixation [5]. An image board is also a very effective tool for users to express their emotions, expectations, and points of view [6]. Brown [7] pointed out that users could obtain more positive emotions for products by purchasing products designed using an image board.

Most designers design an image board by searching and retrieving visual data. Graphical data are beneficial to the design transformation. In the process of design transformation, designers generate image associations through the visual stimulation of pictures before further converting them into a design concept [8]. Therefore, it is necessary to collect huge amounts of image pictures, identify the characteristics of them, sort them into categories accordingly, and provide appropriate names for each category before using them to make an image board. Picture collection is time-consuming and labor-intensive. Furthermore, the pictures are mostly collected subjectively, analyzed, sorted, and named by designers based on their personal experiences and feelings without verifying their conformation to the image that users pursue. Most of the pictures collected are paper based, which occupy a lot of space, and also make it difficult to replace old pictures. Currently, although designers can scan pictures and create folders for them to electronically collect data, it still consumes a lot of time and energy. In addition, the image database created by the designers themselves can have cognitive differences with users in terms of image integration, which is not conducive to a participatory design process among people with various backgrounds.

In order to solve the above-mentioned problems with using an image board, this study takes Lifestyles of Health and Sustainability (LOHAS) as the research object, and through an investigation selects the appropriate image pictures to serve as a picture database. In this preliminary study, a picture database of other lifestyle people is established, for the future, to help designers construct an appropriate image board effectively and quickly for designing products closer to users' positive emotional requirements.

## 2. Literature Review

An image is the mental imaginary of integrating internal subjective emotion and external physical image. An image can be generated in two ways as follows: To be specific, the first way is that the image can be directly obtained from people's own culture, including life experience, social norms, life background, etc.; the second way is that the cognition is conveyed to the brain through sensory stimulation indirectly, such as the imagination after reading or the feeling after appreciating artistic work. Most scholars in different fields believe that the image is composed of cognitive image and affective image [9]. Cognitive image refers to the personal attitude and perception and affective image refers to the personal understanding of image attributes and characteristics [10]. Nagamachi also proposed that an image is abstract and nonspecific, and its imagination is related to users' life experience and cultural background. He regarded an image as an equivalent term to Japanese "kansei" (emotion) [11], although he asserted that the meaning "kansei" was more sophisticated than emotion, and proposed and developed a new ergonomic approach of Kansei engineering.

Image boards provide an environment that stimulates designers' ideas for product development and design [4]. The image board is used as an auxiliary tool in design. Its application range is very wide, and it is commonly used in product design. For example, Taff, Pedell and Wilkinson (2018) reported that designers and the elderly used an image board to discuss the texture, shape, and material of an armrest, and applied it to a simulation of an indoor family environment, to motivate the elderly to choose, and finally purchase the item [12]. It is also used in perfume design [13], clothing design [14],

packaging design [15], and can even be used for research and evaluation of students' learning mood [16]. There are four stages in the design process where image boards can be used, which include the following:

1.  The exploratory stage is used in the stage prior to product design or preliminary design stage with the main purpose of understanding and communicating with users regarding their perception of the products currently available on the market.
2.  The design stage is used in the design conception process to obtain the ideal product concept in the users' mind and stimulate design ideas.
3.  The assessment stage is used to assess developed design concepts to ensure a user-oriented design.
4.  The presentation stage is used to express the image that the completed design is aimed for and is usually used in advertising or marketing.

The three general ways to prepare relevant pictures for image boards [6] are as follows:

1.  By designers who collect pictures relevant to target consumers or the planned product theme mainly from the Internet, newspapers, magazines, advertisements, etc. They collect pictures mainly based on the subjective awareness of designers such as their personal life experience, cultural background, etc., and then position and group the pictures based on their subjective judgement.
2.  By panel discussion, in which, first, pictures are extensively collected by designers according to the lifestyles of the target consumers and expected personality of the product or planned design theme, and then the pictures are positioned, grouped, and named by a panel discussion with expert members.
3.  By both designers and users, in which relevant pictures are extensively collected by designers according to the lifestyle of the target consumers and expected personality of the product or planned design theme, and then they are positioned, grouped, and named by target users based on the same principles.

A mood board is a basic tool used in the design industry, which promotes creative and innovative thinking and application. In the above three methods, most of the images are collected and produced by the designer. The image board reveals the basic principles related to the working method and problem-solving theory of the designer when designing [17]. However, it is always too subjective, and the expressed image is always too superficial or limited, making it difficult to meet users' real needs. Preparing relevant pictures by panel discussion is more objective than the former, but most of the panel experts are either designers or manufacturers, making it difficult to get close to the image that users pursue. Moreover, with this method, the picture image needs to be discussed in depth and compared repetitively, which is absolutely time-consuming and labor-intensive. In the last method of preparing pictures by both designers and users, although users are responsible for picture positioning and grouping, pictures are collected by designers, and the integration of opinions becomes very difficult due to the perception differences of pictures between users and designers. By contrast, the establishment of the picture database proposed in this study entails the joint efforts of designers and users, from picture collection to image board production.

Advancement in science and technology has provided some tools for making image boards, which has prioritize replacing traditional manual image board production with e-webpage platforms. For example, Lucero and Dima proposed an approach for developing an image board using the MR system [18], which provided the required technology and system concepts to construct a tailored image browser, and also provided designers with an interaction platform for making the image board. To use this system, first, designers were required to collect many image pictures themselves. The system provided the functions of downloading pictures from the Internet and scanning paper pictures to assist building the designers' personal picture database. Designers could use their hands comfortably and flexibly to grab physical images (Figure 1) to make image boards in the design studio environment. This e-webpage platform mainly solved the problem of office chaos caused by handmade image boards and provided an electronic picture database that reduced the problem of taking up a lot of space

with paper pictures [19]. However, the picture collecting, positioning, grouping, naming, and image board constructing with this e-webpage platform still involved designers doing so according to their personal views. Its objectiveness was not verified, and the collected pictures could not necessarily meet the users' needs. At the same time, the platform was only suitable for a single user instead of a transdisciplinary joint design team.

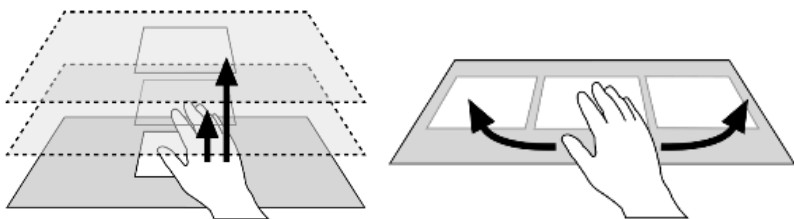

**Figure 1.** Electronic interactive mode of operation proposed by Lucero and Dima (Lucero, 2007).

SmpleBorad Lab is an e-webpage platform that can be used to construct image boards for the following four categories of industry: interior design, landscape, wedding, and clothing and textile. The platform allows users and designers to construct personal image boards to use for stimulating design ideas or communication ideas with others. The picture database of the platform contains about 30,000–50,000 pictures, which are classified according to type of products and industries. The platform software can assist designers with retrieving relevant pictures from a picture database according to the mode of cloth pattern, texture, color pallet, and so on, to use for conveying the design conception (as shown in Figure 2). The completed image boards can be stored based on the four categories above. The classification is based on only four types of industries, and therefore platform users usually cannot quickly obtain appropriate pictures to build their image boards. In addition, they are only designed for four categories of industry, which means a failure of comprehensive coverage for product design that is closely related to varied aspects of people's lives.

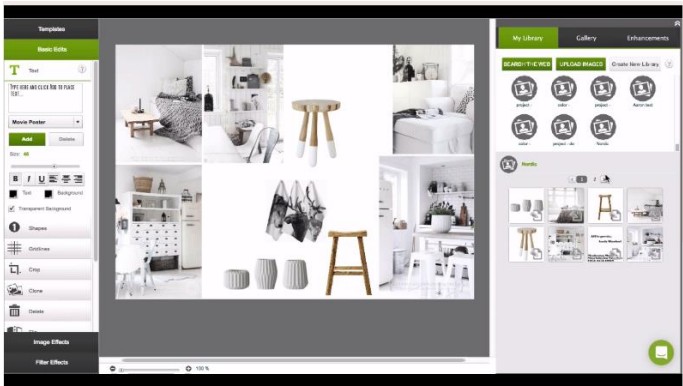

**Figure 2.** SmpleBorad laboratory e-webpage platform.

The MoodShare image board Internet platform is fitted with a drawing function such as the "MoodShare image board Internet platform" (MoodShare, 2019), and search engine linking sources of pictures and video films from such websites as Twitter, Google, Bing, Flickr, Picasa, BigStock, ShutterStock, Youtube, Vimeo, and ColorLovers, etc. Pictures are mainly obtained by entering key nouns or pronouns. For example, by entering the key word "wave", a user can find pictures and videos related to wave, and can quickly drag images, videos, sounds, and color palettes within a few seconds to create an image board (as shown in Figure 3). The picture collection of this platform is based on the picture classification of the individual major websites (mostly by key words), however, users have no idea about how the pictures have been classified, and therefore have difficulty obtaining pictures that are relevant to an ideal image of their target users. Moreover, the pictures are retrieved in this system

by entering a noun or pronoun rather than entering the relevant adjective of a desired image, and the obtained output is multifarious and messy. If these pictures are directly used to make an image board, it is impossible to determine if they meet the image that users pursue. Although we can position and group these pictures before using them to make an image board, it is as time-consuming and labor-intensive as using the traditional image board method. A discussion related to the experience of using these three web pages related to the image board has been provided by users who have used the platform. According to the users' experiences, the pictures have not been tested, and it is sometimes not easy to find pictures that meet their expectations.

The effective use of an image board relies on the appropriateness and objectivity of the picture collection, the simplicity and convenience of the production process, the ability to ensure the participation of multiple people from picture collection to image board production, and no restrictions by region and language. In view of this, this research intends to establish an image picture database and construct an e-webpage platform, both of which involve users' participation to ensure the obtained pictures better reflect LOHAS, and therefore it can be effectively used by designers and relevant industries.

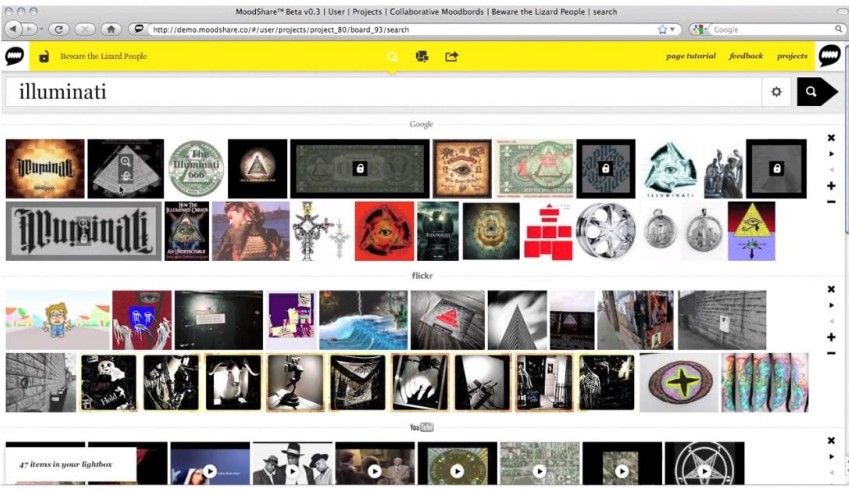

**Figure 3.** Image board production process using Moodshare.

LOHAS was first proposed by Ray in the book The Culture Creatives: How 50 Million People are Changing the World [20]. He proposed that one quarter of the population in the United States and about one third of the population in Europe belong to the LOHAS group and predicted that nearly half of the total population in the USA would be LOHAS in the future. One of three people in Taiwan belong to the LOHAS group, as pointed out by the Eastern Integrated Consumer Profile (E-ICP, 2019), which completed a survey on the consumption behaviors and lifestyle of the consumers in Taiwan per year, since 1998. Pícha and Navrátil [21] pointed out that LOHAS consumers can be identified as a group with specific purchase behaviors. It has been estimated by Australia's LOHAS Consumer Trends Report, that the global value of the LOHAS market would exceed AUD 500 billion. Products whose main targeted consumer group is LOHAS, such as products of Japanese MUJI and Daiso are also very popular in Australia. With the consumers of LOHAS spreading around the world, new and giant business opportunities for LOHAS have been created, as reflected by data, reports, research, and market trends. LOHAS attaches great importance to a healthy life which includes buying organic food; buying local products; being passionate about learning; emphasizing the qualities of actions; leaning towards green, health, and sustainability; and creating a healthy environment to pass on to the next generation [22]. As a result, their families and friends have further influenced the adoption of sustainable and healthier lifestyles [23]. In addition, overall, LOHAS consumers tend to make purchasing decisions that meet their social and environmental responsibility standards [24]. Therefore, LOHAS was taken as the research object in this study.

The semantic differential (SD) scale was first introduced by Osgood, in 1957, to explore the semantic connotative meanings of some abstract concepts [25]. It requires respondents to self-report their feelings on a concept based on a set of opposite semantic adjective pairs. Currently, this method has been widely used in various related fields, and especially in the field of design it is frequently applied to explore the image or emotional feeling of design [26–30]. Thus, the SD scale was also used to explore the image feelings of the collected pictures, in this study.

To statistically analyze data in some systems which are composed of many variables with complex interrelationships, the existing randomness in data can confuse researchers' intuition and increase the complexity of forming a clear concept. In view of the above, Deng (1982) proposed grey relational analysis (GRA) to clarify the main relationships among various factors in a system through a certain method [31,32], to determine the most influential factors, and to check the relevance of two systems. In contrast to regression analysis which has more data and fewer variables, GRA has a very simple and clear calculation process, needs only a small amount of data, and is more flexible in terms of condition limitations than traditional methods. The obtained quantitative results do not produce conclusions in conflict with qualitative analysis. The model assumed is a non-functional sequence model, which can effectively handle discrete data. The analysis steps of GRA include the following: (1) determination of the analysis sequence, (2) data standardization, (3) calculation of grey relational coefficients, (4) calculation of grey relational grade, and (5) ranking of grey relational grade. In the field of design, GRA has also always been used for multicriteria analysis to compare the comprehensive performance among different designs or design concepts. For example, Chen and Chuang applied GRA to explore the aesthetic quality of mobile phones for achieving higher customer satisfaction and showed that GRA is suitable for researching abstract concepts such as society and economic systems [33].

## 3. Method

By taking the LOHAS image as the research object, this research included the following steps: (1) asked representatives of LOHAS and designers to extensively collect pictures and image adjectives relevant to the LOHAS image; (2) invited experts with design experience to choose pictures as stimuli and key adjectives as scales, from the above collection, which are more relevant to the LOHAS image; (3) recruited LOHAS subjects to conduct the SD assessment on the selected stimuli with the selected scales; and (4) selected suitable pictures for expressing the image of LOHAS or the key LOHAS adjectives based on the result of the SD assessment.

### 3.1. Representatives of the LOHAS

This research was focused on LOHAS. The E-ICP Lifestyle Scale, which is a set of questionnaires developed by Dongfang Online in cooperation with the Institute of Business Administration of the National Chengchi University of Taiwan to classify the consumers' lifestyle, was adopted to screen representatives and subjects of LOHAS for this study. First, volunteers were invited, online, to complete the LOHAS questionnaire of the E-ICP Lifestyle Scale. From 68 volunteers, 52 persons belonging to LOHAS were identified. Among them, 16 people (7 males and 9 females), aged mainly above 31–35, and with an educational background of college/university or above, agreed to participate as LOHAS representatives, in this study.

### 3.2. Selected Stimuli of Pictures

There were two methods used to collect the pictures of the LOHAS image. First, the 16 LOHAS representatives were asked to extensively collect pictures, for one week, mainly about food, clothing, residence, travel, education, and recreation related to LOHAS. There were no limitations set on the number of pictures and a total of 325 pictures were collected. Another 5 LOHAS representatives were invited to review and discuss the relevance of these pictures for expressing the LOHAS image via cloud video conference. On the basis of a consensus, 50 pictures were selected for the database. Secondly,

since image boards are mostly built and used by designers, in order to include designers' views in the picture database, 14 designers with more than 1 year of design experience were also invited to, first, collect many pictures relevant to LOHAS and product design. There were no limitations set on the number of pictures collected. The collection lasted one week with a total of 200 pictures collected. Another five designers with more than 5 years of design experience were invited to reduce the quantity of the pictures also via cloud video conference. On the basis of a consensus, 16 pictures were finally selected. Following these two methods, a total of 66 pictures, as shown in Figure 4, were proposed for the picture database and served as the stimuli for the SD assessment.

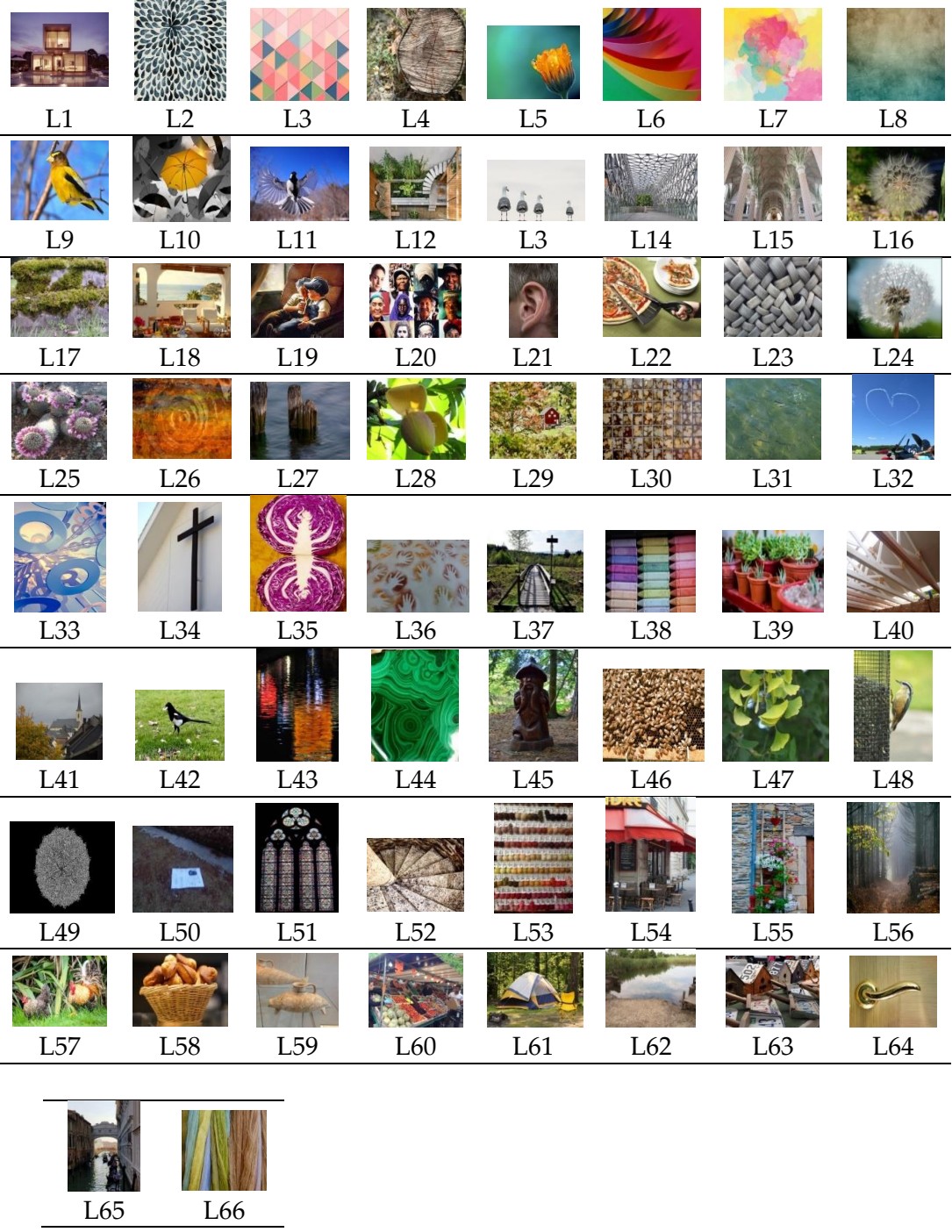

**Figure 4.** Stimuli of pictures for the LOHAS image (L1–L16 were proposed by designers and L17–L66 by the representative of LOHAS).

*3.3. Selected Scales for the SD Assessment*

To expand the range of assessment scales for the LOHAS image, we posted three questionnaires online and asked volunteers of different backgrounds to provide relevant adjectives, over 2 weeks, as shown below:

1.   General respondents were asked to provide adjectives of expected images commonly desired on products; a total of 160 adjectives were collected from 15 general respondents.
2.   Designers were asked to provide adjectives of expected images commonly used by designers on product design; a total of 188 adjectives were obtained from 13 designer respondents.
3.   People belonging to LOHAS were asked to provide adjectives of their expected images on products; a total of 188 adjectives were collected from 22 LOHAS respondents.

The researchers compiled the adjectives which had been collected through the two methods, by combining the adjectives with similar meaning, deleting repetitive adjectives or those with irrelevant connotation, and obtained a total of 148 adjectives which were close to the LOHAS image. Then, five experts with more than five years of design experience were invited to further reduce the number of adjectives together. In consideration of the work load for responding to the questionnaire of the SD assessment, the experts were instructed to select no more than 10 adjectives, and 6 key image adjectives were finally determined, including healthy, environmentally friendly, sustainable, nature, simple, and ecological. These 6 adjectives were adopted as the assessment image scale with a 5-level degree of agreement Likert scale for the SD assessment.

*3.4. SD Assessment*

We recruited subjects of LOHAS to participate in the two-stage SD assessment; in the first stage, they assessed the 50 pictures determined by LOHAS representatives and, in the second stage, they assessed the 16 pictures determined by designers. The subjects were volunteers recruited online, through various APPs (WeChat, Line, and What's up) and community websites (Facebook and Weibo), and screened by the LOHAS questionnaire of the E-ICP Lifestyle Scale. For these two stages, 101 and 118 valid subjects were recruited, respectively. The purpose and instructions for the SD assessment were explained to the subjects, and they were asked to provide their demographic data. Then, they were asked to self-report their feelings on the six key image adjectives using the 5-level degree of agreement Likert scale for each picture, until all pictures were assessed. The assessment was anonymously conducted over 3 months.

*3.5. Data Analysis*

Based on the responses given by each subject, the comprehensive mean score of all respondents for each picture on each image adjective was calculated and ranked. The pictures with higher mean scores were believed to better comply with the LOHAS image. Then, the correlations between pictures and LOHAS were calculated and ranked by grey relational analysis. Again, the pictures with higher grey relational grades were believed to better comply with the LOHAS image. The results of the above two analyses were summarized to recommend suitable pictures for the database of the LOHAS image.

**4. Results**

*4.1. Subject Background*

A total of 219 valid subjects (101 subjects for assessing pictures by LOHAS representatives and 118 subjects for assessing pictures by designers) were recruited including 122 males and 97 females, most of whom were aged 25–45 (59.82%). According to the demographic data that was provided, the subjects had independent economic ability, mostly associated with service industries, and had an educational background of college or junior college (including a master's degree or above, 56.62%). In terms of age distribution and career, the group had a life attitude of continuous learning.

*4.2. Semantic Difference Assessment Results*

The degree of agreement of each subject's SD assessment on each adjective using a 5-level Likert scale for each picture was, first, converted into a score of 1–5; then, the average scores of the subjects for each picture on each of the six key adjectives were calculated, as shown in the Columns 3–8 of Table 1. Next, the grand average values of the six adjectives for each picture were calculated, as shown in Columns 9–10 of Table 1. Ranking the relevance of the pictures to the LOHAS image (Column 1 of Table 1) was completed based on the grand average values. Finally, the number of the image adjectives with an average value greater than four (corresponding to the degree level of "agree"or above on the 5-level degree of agreement Likert scale) for each picture was counted, as shown in Column 10 of Table 1.

Two criteria were adopted for recommending relevant pictures based on the above results. The first criterion was the grand average value (Columns 9 in Table 1) of a picture higher than four and there were 16 pictures that meet the criterion including L12, L11, L56, L28, L46, L47, L29, L55, L31, L4, L16, L62, L1, L27, L35, and L60 in order of grand average values, as shown in Figures 5 and 6. The second criterion was pictures with the total number of adjectives average value higher than four (Columns 10 in Table 1) equal to six (all adjective average values >4) and there were 12 pictures that meet the criterion including L12, L11, L56, L28, L46, L47, L29, L55, L31, L4, L16, and L1 in order of grand average values, as shown in Figure 5. These 12 pictures screened by the second criterion certainly also met the first criterion, and thus they were regarded as the most recommended pictures for the LOHAS image, by this study. The other four pictures including L62, L27, L35, and L60, as shown in Figure 6, which only met the first criterion and not the second criterion, were regarded as recommended pictures for the LOHAS image, by this study.

Focused on the individual image adjective, with the criterion of an average value higher than four, there are 20 pictures for healthy, 18 for environmentally friendly, 21 for sustainable, 21 for nature, 19 for simple, and 19 for ecological are recommend, as shown in Table 2.

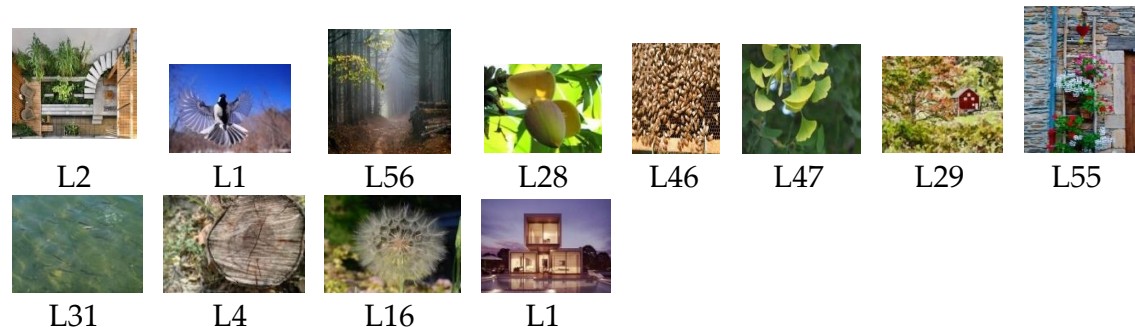

**Figure 5.** Most recommended pictures with all adjective average values >4.

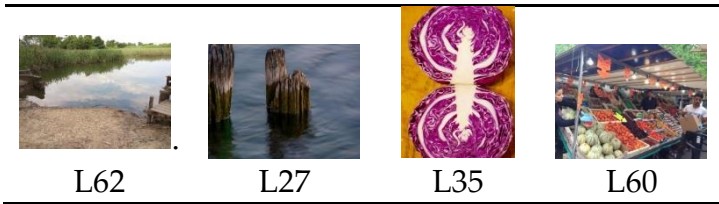

**Figure 6.** Recommended pictures with a grand average value >4, but not all adjective average values >4.

**Table 1.** SD assessment results.

| Rank | Picture Number | Grand Average | Adjectives Average Number of 4 or More | Rank | Picture Number | Grand Average | Adjectives Average Number of 4 or More | Rank | Picture Number | Grand Average | Adjectives Average Number of 4 or More |
|---|---|---|---|---|---|---|---|---|---|---|---|
| 1 | L12 | 4.67 | 6 | 23 | L41 | 3.88 | 0 | 45 | L65 | 3.23 | 0 |
| 2 | L11 | 4.62 | 6 | 24 | L64 | 3.77 | 1 | 46 | L2 | 3.11 | 0 |
| 3 | L56 | 4.60 | 6 | 25 | L36 | 3.75 | 0 | 47 | L63 | 3.01 | 0 |
| 4 | L28 | 4.59 | 6 | 26 | L45 | 3.70 | 0 | 48 | L10 | 2.99 | 1 |
| 5 | L46 | 4.59 | 6 | 27 | L58 | 3.68 | 0 | 49 | L52 | 2.98 | 0 |
| 6 | L47 | 4.58 | 6 | 28 | L32 | 3.67 | 2 | 50 | L19 | 2.98 | 0 |
| 7 | L29 | 4.56 | 6 | 29 | L61 | 3.65 | 0 | 51 | L21 | 2.97 | 0 |
| 8 | L55 | 4.55 | 6 | 30 | L26 | 3.62 | 0 | 52 | L37 | 2.97 | 0 |
| 9 | L31 | 4.49 | 6 | 31 | L42 | 3.51 | 0 | 53 | L59 | 2.95 | 0 |
| 10 | L4 | 4.35 | 6 | 32 | L24 | 3.51 | 0 | 54 | L20 | 2.93 | 0 |
| 11 | L16 | 4.35 | 6 | 33 | L-6 | 3.49 | 1 | 55 | L3 | 2.92 | 0 |
| 12 | L62 | 4.34 | 5 | 34 | L34 | 3.49 | 1 | 56 | L51 | 2.90 | 0 |
| 13 | L1 | 4.27 | 6 | 35 | L25 | 3.48 | 0 | 57 | L4 | 2.87 | 0 |
| 14 | L27 | 4.27 | 5 | 36 | L15 | 3.47 | 1 | 58 | L7 | 2.86 | 0 |
| 15 | L35 | 4.22 | 4 | 37 | L44 | 3.46 | 3 | 59 | L8 | 2.80 | 0 |
| 16 | L60 | 4.19 | 3 | 38 | L57 | 3.46 | 0 | 60 | L6 | 2.78 | 0 |
| 17 | L49 | 3.96 | 2 | 39 | L0 | 3.46 | 1 | 61 | L5 | 2.75 | 0 |
| 18 | L40 | 3.94 | 1 | 40 | L33 | 3.43 | 1 | 62 | L9 | 2.68 | 0 |
| 19 | L53 | 3.93 | 2 | 41 | L17 | 3.40 | 0 | 63 | L30 | 2.64 | 0 |
| 20 | L43 | 3.92 | 2 | 42 | L39 | 3.35 | 2 | 64 | L22 | 2.49 | 0 |
| 21 | L38 | 3.91 | 0 | 43 | L18 | 3.28 | 0 | 65 | L23 | 2.39 | 0 |
| 22 | L54 | 3.88 | 1 | 44 | L48 | 3.24 | 0 | 66 | L13 | 2.22 | 0 |

**Table 2.** Recommend pictures for individual image adjectives.

| Adjective | Picture Number |
|---|---|
| Healthy | L12, L54, L60, L43, L28, L46, L47, L55, L62, L11, L29, L32, L34, L31, L56, L16, L4, L49, L1, L44 (Total 20) |
| Environmentally friendly | L28, L12, L56, L11, L47, L29, L31, L46, L55, L10, L16, L1, L4, L66, L44, L27, L53, L39 (Total 18) |
| Sustainable | L35, L12, L15, L60, L27, L11, L62, L56, L29, L47, L46, L28, L55, L16, L31, L4, L40, L32, L1, L49, L33 (Total 21) |
| Nature | L35, L4, L56, L62, L28, L12, L11, L46, L47, L55, L29, L27, L31, L24, L25, L1, L39, L16, L43, L64, L66 (Total 21) |
| Simple | L12, L1, L56, L11, L46, L32, L61, L55, L62, L31, L47, L29, L28, L7, L4, L16, L27, L43, L53 (Total 19) |
| Ecological | L11, L62, L29, L46, L42, L28, L34, L56, L47, L55, L27, L16, L12, L31, L4, L44, L1, L35, L50 (Total 19) |

*4.3. Grey Relational Analysis*

The grey relational analysis (GRA) contains a reference series (also called parent series) that reflects the characteristics of the system behavior, and a compared series. The calculation formula and steps are as follows:

**Reference** $x_0(k) = \{(x_0(1), x_0(2), \ldots, x_0(n)\}$, k = 1,2,3, … , n.

**Compared Series** $x_i(k) = \{(x_i(1), x_i(2), \ldots, x_i(n)\}$ i = 1,2,3, … , m.

If m series is compared by n attributes, m = 66 pictures and n = 6 adjectives, in this study.

**Analysis Step 1** Select the appropriate reference series. In this research, the maximum value, 5, of the SD assessment for the six image adjectives is taken as the reference series.

**Analysis Step 2** For data normalization, normalize the values shown in Columns 3 to 8 of Table 1. The most common methods are min-max standardization and z-score standardization; there is no fixed use standard. In this research, the z-score standardization was adopted to let the processed data conform to the standard normal distribution, which means the average value is 0 and the standard deviation is 1.

**Analysis Step 3** Calculate the grey relational coefficient as follows:

$$r(x_0(k),\ x_i(k)) = \frac{\Delta min + \zeta \Delta max}{\Delta oi(k) + \zeta \Delta max} \tag{1}$$

where r($x_0(k)$, $x_i(k)$) are the grey relational coefficients of the $k^{th}$ attribute (adjective) and the $i^{th}$ series (picture), and $\Delta_{0i}(k) = |x_0(k) - x_i(k)|$ is the absolute difference between $x_0(k)$ and $x_i(k)$. $\Delta_{min}$ and $\Delta_{max}$ are, respectively, the minimum and maximum values of the absolute difference from the reference series of the compared series at each point. $\zeta$ is the distinguish coefficient, between 0 and 1, for adjusting distinguish resolution. The value of 0.5 was adopted in this study, as in general studies. The grey relational coefficient of each picture on each adjective was calculated according to the above formula and the results are shown in Columns 3 to 8 of Table 3.

**Analysis Step 4** Calculate the grey relational grade from the grey relational coefficients. The commonly used calculation methods of grey relational grade are average value method and weighted method. The average value method, assuming equal weights of all factors, was used for this research. According to the following formula, the calculated grey relational grade of each picture is shown in Columns 9 of Table 3:

$$r(x_0,\ x_i)\ \frac{1}{n} \sum_{k=1}^{n} r(x_0(k),\ x_i(k)) \tag{2}$$

**Analysis Step 5** Determine the grey relational order. Rank pictures according to the grey relational grades, as shown in Column 1 of Table 3.

**Analysis Step 6** Select the suitable series. Select the recommended pictures according to the results of the GRA.

Again, two criteria were adopted to recommend relevant pictures based on the results of the GRA. The first criterion is the grey relational grade (Column 9 in Table 3) of a picture higher than 0.8 and there are 13 pictures that meet the criterion including L12, L11, L56, L46, L28, L47, L29, L31, L55, L62, L16, L4, and L27 in order of grey relational grade, as is shown in Figures 7 and 8. The second criterion is pictures with the total number of grey relational coefficients of adjectives higher than 0.8 (shown in Column 10 of Table 3) equal to 6 (all grey relational coefficients of adjectives >0.8) and there are nine pictures that meet the criterion including L12, L11, L6, L46, L28, L47, L29, L31, and L55 in order of grey relational grade, as shown in Figure 7. The nine pictures screened by the second criterion certainly also meet the first criterion, thus, they are regarded as the most recommended pictures for the LOHAS image, here. The other four pictures including L62, L16, L4, and L27, as shown in Figure 8, only meet the first criterion but not the second one, are regarded as recommended pictures for the LOHAS image.

Considering the individual image adjective, with the criterion of grey relational coefficient higher than 0.8, there are 16 pictures for healthy, 12 for environmentally friendly, 17 for sustainable, 14 for nature, 15 for simple, and 14 for ecological recommended, respectively, as shown in Table 4.

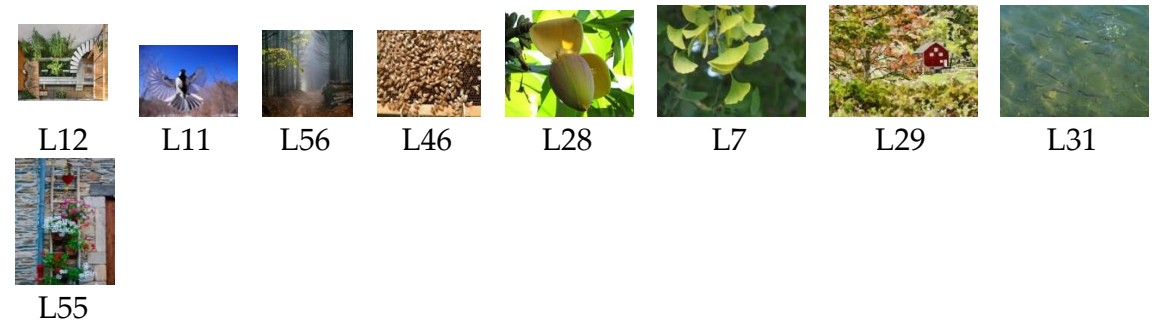

L12　　　L11　　　L56　　　L46　　　L28　　　L7　　　L29　　　L31

L55

**Figure 7.** Most recommended pictures with all adjective grey relational coefficients >0.8.

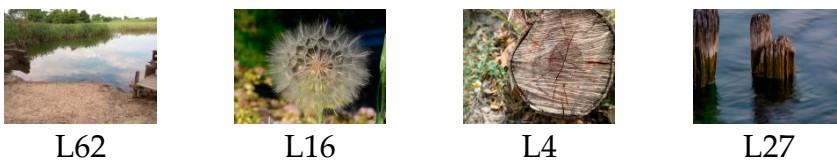

L62　　　　　　L16　　　　　　L4　　　　　　L27

**Figure 8.** Recommended pictures with the grey relational grade >0.8, but not all adjective grey relational coefficients >0.8.

**Table 3.** Result of the GRA.

| Rank | Picture Number | Healthy | Environmentally Friendly | Sustainable | Nature | Simple | Ecological | Grey Relational Grade | The Number of Adjectives with a Grey Relational Coefficient Above 0.8 |
|------|---------|---------|------------------|-------------|--------|--------|------------|------------------|------------------------------------------------------------|
| 1 | L12 | 0.92 | 0.89 | 0.91 | 0.88 | 0.94 | 0.84 | 0.8967 | 6 |
| 2 | L11 | 0.89 | 0.89 | 0.89 | 0.88 | 0.90 | 0.91 | 0.8933 | 6 |
| 3 | L56 | 0.85 | 0.89 | 0.88 | 0.90 | 0.90 | 0.88 | 0.8833 | 6 |
| 4 | L46 | 0.89 | 0.87 | 0.87 | 0.88 | 0.88 | 0.89 | 0.8800 | 6 |
| 5 | L28 | 0.89 | 0.89 | 0.86 | 0.89 | 0.86 | 0.88 | 0.8783 | 6 |
| 6 | L47 | 0.89 | 0.88 | 0.87 | 0.88 | 0.86 | 0.87 | 0.8750 | 6 |
| 7 | L29 | 0.87 | 0.87 | 0.88 | 0.86 | 0.87 | 0.89 | 0.8733 | 6 |
| 8 | L31 | 0.87 | 0.88 | 0.86 | 0.85 | 0.89 | 0.87 | 0.8700 | 6 |
| 9 | L55 | 0.88 | 0.86 | 0.86 | 0.87 | 0.88 | 0.86 | 0.8683 | 6 |
| 10 | L62 | 0.88 | 0.54 | 0.89 | 0.89 | 0.87 | 0.99 | 0.8435 | 5 |
| 11 | L16 | 0.82 | 0.82 | 0.85 | 0.73 | 0.76 | 0.85 | 0.8050 | 4 |
| 12 | L4 | 0.77 | 0.77 | 0.80 | 0.90 | 0.80 | 0.77 | 0.8017 | 3 |
| 13 | L27 | 0.70 | 0.74 | 0.9 | 0.85 | 0.75 | 0.86 | 0.8000 | 3 |
| 14 | L1 | 0.73 | 0.84 | 0.77 | 0.78 | 0.92 | 0.74 | 0.7967 | 2 |
| 15 | L35 | 0.71 | 0.68 | 0.98 | 0.97 | 0.68 | 0.75 | 0.7950 | 2 |
| 16 | L60 | 0.90 | 0.73 | 0.91 | 0.74 | 0.71 | 0.72 | 0.7850 | 2 |
| 17 | L43 | 0.91 | 0.72 | 0.73 | 0.75 | 0.75 | 0.52 | 0.7300 | 1 |
| 18 | L40 | 0.73 | 0.69 | 0.83 | 0.70 | 0.70 | 0.72 | 0.7283 | 1 |
| 19 | L49 | 0.78 | 0.72 | 0.74 | 0.71 | 0.71 | 0.71 | 0.7283 | 0 |
| 20 | L54 | 0.92 | 0.71 | 0.72 | 0.7 | 0.71 | 0.53 | 0.7150 | 1 |
| 21 | L53 | 0.70 | 0.74 | 0.70 | 0.72 | 0.72 | 0.70 | 0.7133 | 0 |
| 22 | L38 | 0.71 | 0.72 | 0.71 | 0.71 | 0.73 | 0.68 | 0.7100 | 0 |
| 23 | L41 | 0.71 | 0.69 | 0.72 | 0.73 | 0.7 | 0.71 | 0.7100 | 0 |

**Table 3.** *Cont.*

| Rank | Picture Number | Healthy | Environmentally Friendly | Sustainable | Nature | Simple | Ecological | Grey Relational Grade | The Number of Adjectives with a Grey Relational Coefficient Above 0.8 |
|------|------|------|------|------|------|------|------|------|------|
| 24 | L36 | 0.72 | 0.71 | 0.72 | 0.72 | 0.72 | 0.53 | 0.6867 | 0 |
| 25 | L64 | 0.54 | 0.70 | 0.72 | 0.73 | 0.71 | 0.72 | 0.6867 | 0 |
| 26 | L32 | 0.87 | 0.57 | 0.77 | 0.51 | 0.88 | 0.51 | 0.6850 | 2 |
| 27 | L45 | 0.70 | 0.51 | 0.72 | 0.72 | 0.70 | 0.72 | 0.6783 | 0 |
| 28 | L61 | 0.69 | 0.54 | 0.71 | 0.53 | 0.88 | 0.70 | 0.6750 | 1 |
| 29 | L26 | 0.71 | 0.70 | 0.55 | 0.72 | 0.69 | 0.57 | 0.6567 | 0 |
| 30 | L42 | 0.53 | 0.72 | 0.72 | 0.54 | 0.52 | 0.89 | 0.6533 | 1 |
| 31 | L34 | 0.88 | 0.53 | 0.53 | 0.54 | 0.55 | 0.88 | 0.6517 | 2 |
| 32 | L24 | 0.54 | 0.55 | 0.56 | 0.85 | 0.70 | 0.70 | 0.6500 | 1 |
| 33 | L5 | 0.48 | 0.68 | 0.94 | 0.55 | 0.67 | 0.53 | 0.6417 | 1 |
| 34 | L25 | 0.54 | 0.54 | 0.55 | 0.78 | 0.71 | 0.71 | 0.6383 | 0 |
| 35 | L50 | 0.71 | 0.54 | 0.74 | 0.55 | 0.54 | 0.75 | 0.6383 | 0 |
| 36 | L58 | 0.63 | 0.65 | 0.66 | 0.63 | 0.63 | 0.63 | 0.6383 | 0 |
| 37 | L66 | 0.54 | 0.77 | 0.72 | 0.73 | 0.54 | 0.53 | 0.6383 | 0 |
| 38 | L44 | 0.74 | 0.75 | 0.53 | 0.52 | 0.51 | 0.77 | 0.6367 | 0 |
| 39 | L57 | 0.70 | 0.52 | 0.54 | 0.54 | 0.74 | 0.75 | 0.6317 | 0 |
| 40 | L33 | 0.54 | 0.71 | 0.72 | 0.53 | 0.53 | 0.71 | 0.6233 | 0 |
| 41 | L17 | 0.66 | 0.57 | 0.55 | 0.70 | 0.53 | 0.72 | 0.6217 | 0 |
| 42 | L39 | 0.55 | 0.74 | 0.55 | 0.76 | 0.55 | 0.54 | 0.6150 | 0 |
| 43 | L18 | 0.53 | 0.55 | 0.55 | 0.70 | 0.73 | 0.53 | 0.5983 | 0 |
| 44 | L65 | 0.53 | 0.53 | 0.56 | 0.52 | 0.71 | 0.72 | 0.5950 | 0 |
| 45 | L48 | 0.52 | 0.54 | 0.53 | 0.54 | 0.73 | 0.71 | 0.5950 | 0 |
| 46 | L2 | 0.62 | 0.68 | 0.48 | 0.49 | 0.67 | 0.50 | 0.5733 | 0 |

**Table 3.** *Cont.*

| Rank | Picture Number | Healthy | Environmentally Friendly | Sustainable | Nature | Simple | Ecological | Grey Relational Grade | The Number of Adjectives with a Grey Relational Coefficient Above 0.8 |
|------|------|------|------|------|------|------|------|------|------|
| 47 | L63 | 0.55 | 0.55 | 0.55 | 0.54 | 0.54 | 0.56 | 0.5483 | 0 |
| 48 | L9 | 0.52 | 0.56 | 0.54 | 0.55 | 0.54 | 0.56 | 0.5450 | 0 |
| 49 | L0 | 0.48 | 0.85 | 0.44 | 0.49 | 0.50 | 0.51 | 0.5450 | 0 |
| 50 | L37 | 0.56 | 0.55 | 0.54 | 0.54 | 0.54 | 0.53 | 0.5433 | 0 |
| 51 | L21 | 0.54 | 0.54 | 0.53 | 0.58 | 0.52 | 0.55 | 0.5433 | 0 |
| 52 | L52 | 0.55 | 0.54 | 0.57 | 0.54 | 0.54 | 0.52 | 0.5433 | 0 |
| 53 | L59 | 0.52 | 0.54 | 0.57 | 0.53 | 0.54 | 0.56 | 0.5433 | 0 |
| 54 | L20 | 0.54 | 0.52 | 0.55 | 0.53 | 0.54 | 0.56 | 0.5400 | 0 |
| 55 | L51 | 0.50 | 0.54 | 0.53 | 0.55 | 0.54 | 0.53 | 0.5317 | 0 |
| 56 | L14 | 0.47 | 0.5 | 0.63 | 0.46 | 0.61 | 0.52 | 0.5317 | 0 |
| 57 | L7 | 0.51 | 0.46 | 0.45 | 0.47 | 0.85 | 0.44 | 0.5300 | 0 |
| 58 | L3 | 0.49 | 0.49 | 0.5 | 0.5 | 0.67 | 0.48 | 0.5217 | 0 |
| 59 | L8 | 0.50 | 0.50 | 0.50 | 0.64 | 0.46 | 0.47 | 0.5117 | 0 |
| 60 | L5 | 0.50 | 0.48 | 0.51 | 0.50 | 0.50 | 0.50 | 0.4983 | 0 |
| 61 | L6 | 0.49 | 0.50 | 0.49 | 0.50 | 0.49 | 0.49 | 0.4933 | 0 |
| 62 | L30 | 0.50 | 0.47 | 0.42 | 0.55 | 0.58 | 0.43 | 0.4917 | 0 |
| 63 | L9 | 0.50 | 0.50 | 0.44 | 0.49 | 0.54 | 0.47 | 0.4900 | 0 |
| 64 | L22 | 0.48 | 0.45 | 0.46 | 0.42 | 0.55 | 0.54 | 0.4833 | 0 |
| 65 | L23 | 0.43 | 0.42 | 0.44 | 0.42 | 0.55 | 0.51 | 0.4617 | 0 |
| 66 | L13 | 0.44 | 0.42 | 0.43 | 0.41 | 0.43 | 0.44 | 0.4283 | 0 |

**Table 4.** Pictures with the grey relational coefficient of image adjectives >0.8.

| Adjective | Picture Number |
|---|---|
| Healthy | L12, L54, L43, L60, L11, L28, L46, L47, L34, L55, L62, L29, L31, L32, L56, L16 (Total 16) |
| Environmentally friendly | L11, L12, L28, L56, L31, L47, L29, L46, L55, L10, L1, L16 (Total 12) |
| Sustainable | L35, L15, L12, L60, L27, L11, L62, L29, L56, L46, L47, L28, L31, L55, L16, L40, L4 (Total 17) |
| Nature | L35, L4, L56, L28, L62, L11, L12, L46, L47, L55, L29, L24, L27, L31 (Total 14) |
| Simple | L12, L1, L11, L56, L31, L32, L46, L55, L61, L29, L62, L28, L47, L7, L4 (Total 15) |
| Ecological | L62, L11, L29, L42, L46, L28, L34, L56, L31, L47, L27, L55, L16, L12 (Total 14) |

## 5. Discussion

According to the demographic data of volunteers and subjects recruited online for this study, there were more males than females, which reflects that males could be more active in the cyber world or more passionate to be volunteers for survey activities. The valid 219 LOHAS subjects selected from 290 volunteers demonstrates that 75.51% of the volunteers can be classified as LOHAS consumers in this study. This, as well as the fact that most of our subjects are well educated people with independent and strong economic ability, implies that huge commercial opportunities exist by taking LOHAS as the target consumer group. In addition, because the number of male subjects selected from the volunteers (57.24%) is higher than that of female subjects (42.76%), it indicates that males are more LOHAS leaning than females.

Among the 16 recommended pictures by the analysis of averaged SD assessments (as shown in Figures 5 and 6), which also included all 13 recommended pictures by GRA, five pictures were selected from the 16 designers' proposed pictures (L1–L16) and 12 pictures selected from the 50 proposed pictures by LOHAS representatives (L17–L66). The rate of designers' proposed pictures selected was 0.31 (5/16) as compared with that of the LOHAS representatives with the value of 0.22 (11/50), therefore, we conclude that the proposed pictures by designers are more suitable for expressing the LOHAS image than those by LOHAS representatives. However, if we examine the nine most recommended pictures by GRA (as shown in Figure 7), which are also e most recommended by the analysis of average value, we find two pictures are from the proposed pictures by the designers, whereas seven pictures are from those by the LOHAS representatives. The selecting rate of proposed pictures by designers 0.125 (2/16) is slightly smaller that of the LOHAS representatives with the value of 0.14 (7/50). Thus, both proposed pictures by designers and by LOHAS representatives can be equally effective for retrieving pictures to express the LOHAS image on an image board.

By examining the contents of the recommended pictures, we find that most of pictures are about natural scenes, animals, plants, fruit and static objects, which imply the characteristics of LOHAS, such as respecting and complying with nature, the importance attached to an organic living environment, and distaste of artificial objects. They prefer green, yellow green, and inherent color of objects. The characteristics of the LOHAS identified above can be applied to further expand the picture database for the LOHAS image board if needed.

The results of picture recommendation by two different analysis approaches, i.e., the analysis of averaged SD assessments and by the GRA, are compared and we summarize the cross relationship of these results in Table 5. In this table, the three columns denote the three levels of recommendation, i.e., most recommended, recommended, and proposed (not recommended), for the 66 pictures been classified by the analysis of averaged SD assessments, whereas the three rows denote those by the GRA. Then, each picture can be filled into one of the nine cells of this table according to its corresponding recommendation classification by the two approaches. From this table, on the one hand, we find that all of the 13 recommended pictures by the GRA are a subset of the 16 recommended pictures by the analysis of averaged SD assessments and all of the nine most recommended pictures by the GRA

are also a subset of the 12 most recommended pictures by the analysis of averaged SD assessments. On the other hand, pictures L4, L16, and L1 are most recommended by the analysis of averaged SD assessments, but are only recommended by the GRA (L4, L16) or not recommended (L1) by the GRA. Thus, we conclude that the screening criterion of the GRA, adopted in this study for picture recommendation, is stricter than that of analysis of averaged SD assessments.

To integrate the results of picture recommendation by two different analysis approaches, we develop a finer recommendation strategy with five recommendation levels based on Table 5. The first recommended pictures are the nine pictures most recommended by both the analysis of averaged SD assessments and the GRA s including L12, L11, L6, L46, L28, L47, L29, L31, and L55, as shown in Table 5. The second recommended pictures are pictures most recommended by one approach but only recommended by another approach, including L4 and L16, which are most recommended by the analysis of averaged SD assessments but are only recommended by GRA. The third recommended pictures are pictures recommended but not most recommended by both approaches (L62 and L27 shown in Table 5), or pictures most recommended by one approach but not recommended at all by another approach, as shown in Table 5, L1 is most recommended by the analysis of averaged SD assessments but not recommended by GRA. The fourth recommended pictures are pictures recommended by one approach but not recommended by another approach, as shown in Table 5, L60 and L35 are recommended by the analysis of averaged SD assessments but not recommended by GRA. The other 50 pictures are not recommended by both approaches which are regarded as the fifth recommended pictures or proposed pictures in this study. In summary, there are nine first recommended pictures, two second recommended pictures, three third recommended pictures, two fourth recommended pictures, and 50 proposed pictures suggested by this study for the LOHAS image according to the above recommendation strategy.

In the same manner, we can compare and integrate the results of picture recommendations by two different analysis approaches for individual images of the six key adjectives. For example, we compile Table 6 to describe the cross relationship of the pictures recommended by the analysis of averaged SD assessments and by the GRA for healthy images. Again, from this table we can conclude the screen criterion of the GRA, adopted in this study for picture recommendation, is stricter than that of analysis of averaged SD assessments, since the 16 recommended pictures by the GRA are the subset of the 20 recommended pictures by the analysis of averaged SD assessments. By integrating both results of picture recommendation, we can further classify recommended pictures into two levels, i.e., the most recommended pictures which are the 16 pictures recommended by both the analysis of averaged SD assessments and GRA, and the recommended pictures which are the four pictures recommended by the analysis of averaged SD assessments but not by GRA, as shown in Table 6.

**Table 5.** The cross relationship of recommended pictures for LOHAS image by the two different analyses.

| By GRA \ By Mean Values | Most Recommended | Recommended | Proposed |
|---|---|---|---|
| Most recommended | First recommended L11, L12, L28, L29, L31, L46, L47, L55, L56 (Total 9) | Second recommended (Total 0) | Third recommended (Total 0) |
| Recommended | Second recommended L4, L16 (Total 2) | Third recommended L62, L27 (Total 2) | Forth recommended (Total 0) |
| Proposed | Third recommended L1 (Total 1) | Forth recommended L60, L35 (Total 2) | L2, L3, L5, L6, L7, L8, L9, L10, L13, L14, L15, L17, L18, L19, L20, L21, L22, L23, L24, L25, L26, L30, L32, L33, L34, L36, L37, L38, L39, L40, L41, L42, L43, L44, L45, L48, L49, L50, L51, L52, L53, L54, L57, L58, L59, L61, L63, L64, L65, L66 (Total 50) |

**Table 6.** The cross relationship of recommended pictures for healthy images by the two different analyses.

| By GRA \ By Mean Value | Recommended | Proposed |
|---|---|---|
| Recommended | Most recommended<br>L11, L12, L16, L28, L29, L31, L32, L34, L43, L46, L47, L54, L55, L56, L60, L62 (Total 16) | Recommended<br>(Total 0) |
| Proposed | Recommended<br>L1, L4, L44, L49 (Total 4) | L2, L3, L5, L6, L7, L8, L9, L10, L13, L14, L15, L17, L18, L19, L20, L21, L22, L23, L24, L25, L26, L27, L30, L33, L35, L36, L37, L38, L39, L40, L41, L42, L45, L48, L50, L51, L52, L53, L57, L58, L59, L61, L63, L64, L65, L66 (Total 46) |

## 6. Conclusions

To establish a user-driven picture database for the LOHAS image, this study recruited LOHAS representatives to collect and determine 50 pictures that expressed the LOHAS image. To include the designers' view in the database, another 16 pictures were added to the pool. The design experts were also asked to identify six key adjectives that expressed the LOHAS image, from a pool collected by general respondents, LOHAS representatives, and designers. By adopting these six key adjectives as an assessing scale, 219 LOHAS subjects were recruited to assess the total 66 pictures in the SD assessment survey. Through the analysis of the averaged SD assessment, the GRA, or by integrating the results of these two analyses, relevant pictures were recommended with various recommending levels that expressed the LOHAS image or expressed the image of individual key adjectives of LOHAS, respectively.

The results of this study have established a picture database to be operated on an e-webpage platform, which is we plan to construct to help designers or related people effectively create an image board of LOHAS. There are three operation modes to be equipped in the e-webpage platform. In the first mode, the users can retrieve pictures for expressing the LOHAS image by selecting the criterion of the averaged SD assessment, the criterion of the GRA, or the integrated criterion, then the corresponding recommended pictures with various recommending levels appear on screen for further operation. In the second mode, the users can retrieve pictures for expressing the image of individual key adjectives of LOHAS, in the same manner. In the third mode, the users can call out all 66 proposed pictures on the screen and select the demanded one from them, then, the parameters of this picture relevant to LOHAS image, including the average scores/ranks of the six key adjectives, the grand average scores/ranks of LOHAS image, the grey relational coefficients/ranks of the six key adjectives, the grey relational grade/rank of LOHAS image, and the recommendation levels by different criteria, are displayed.

To establish a more comprehensive picture database for image board construction, we plan to explore pictures suitable for expressing images of other types of lifestyle and their corresponding adjectives. Additionally, while the equal weights of six key adjectives were assumed for calculating the grand averaged value of the SD assessment and the grey relational grade for each picture in this study, we are trying to test whether adopting different weights of adjectives determined by an appropriate method, such as AHP or entropy, would select pictures to better meet the expected image of consumers. We expect the very tool developed can be effectively used in new product R&D and product marketing of enterprises, and for cultivating professional designers within the field of education.

**Author Contributions:** Conceptualization, P.-J.L. and M.-C.C.; methodology, P.-J.L. and M.-C.C.; software, P.-J.L.; validation, P.-J.L. and M.-C.C.; formal analysis, P.-J.L.; investigation, P.-J.L.; resources, P.-J.L. and M.-C.C.; data curation, P.-J.L.; writing—original draft preparation, P.-J.L.; writing—review and editing, M.-C.C. and C.-C.H.; visualization, P.-J.L.; supervision, M.-C.C. and C.-C.H.; project administration, P.-J.L., M.-C.C. and C.-C.H.; funding acquisition, P.-J.L. and M.-C.C. All authors have read and agreed to the published version of the manuscript.

**Funding:** Ministry of Science and Technology, Taiwan. (NSC102-2221-E-009-104-MY3). The APC was funded by M.-C.C. and P.-J.L.

**Conflicts of Interest:** The authors declare no conflict of interest.

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
