# Peer review of "Establishing Picture Databases for Image Boards: An Example for Lifestyles of Health and Sustainability Images"

_designs, 2020_

Round 1
Reviewer 1 Report
The images board is an effective tool to inspire the designer’s ideas and to help them to understand users better. So the purpose of this research to establish a picture database for images board is valuable. There are some questions and suggestion as
follows:
1. What is the main contribution of this paper a database or a method to establish a database or making a images board? If it is to establish a database, is there the number of samples enough to inspire designers? If a method, didn't find any execu table framework in the conclusion.
2. In the literature review, to compare the three general ways of preparing relevant pictures for image board, should combine the feedback from other scholars instead of analysis just from the author's opinion.
3. As for t he review of LOHAS, it mainly explains its importance but lacks the analysis of characteristics. This part is important for the establishment of evaluation criteria.
4. The SD assessment result does not need to be displayed in full, but the data that have s ignificance are reserved.
5. In conclusion, it is not enough to reflect males are more active in the cyber world, more passionate and toward LOHAS just based on the volunteer with 122 males and 97 females.
Author Response
Designs
June 23, 2020
Dear Editor and Reviewers,
Many thanks for giving me an opportunity to revise our manuscript entitled “Establishing Picture Database for Images Board - An Example for LOHAS Image " (designs-830276). I appreciated the editor’s and reviewers’ positive and constructive comments and suggestions concerning my manuscript. These comments were all very helpful for revising and improving my paper, and also offered guidance in regard to the significance of our research.
The reviewers pointed out nine major problems:
Reviewer 1
- What is the main contribution of this paper a database or a method to establish a database or making a images board?
- In the literature review, to compare the three general ways of preparing relevant pictures for image board, should combine the feedback from other scholars instead of analysis just from the author's opinion.
- As for t he review of LOHAS, it mainly explains its importance but lacks the analysis of characteristics.
- The SD assessment result does not need to be displayed in full, but the data that have s ignificance are reserved.
- In conclusion, it is not enough to reflect males are more active in the cyber world, more passionate and toward LOHAS just based on the volunteer with 122 males and 97 females.
- Herewith please find the attached revised version. Modified text is displayed in red. The responses to the reviewers’ comments are appended below. If you have further comments, please let me know and we will do my best to make any necessary improvements.
Sincerely,
Peng-Jyun Liu
Institute of Applied Arts,National Chiao Tung University &
Department of Creative Product Design, Asia University

Reviewer 2 Report
I have provided a comprehensive overview of the editing requirements to meet a satisfactory level of English comprehension. The 4-page revision document attached also contains more conceptual / high level comments that I feel should be addressed. With these changes I would be happy to support publication.
Conceptual / High level Comments
I have provided a very comprehensive overview of the editing requirements to meet a satisfactory level of English comprehension.
The following are more conceptual / high level comments that I feel should be addressed. With these changes I would be willing to support publication.
104-106. "Among the above three methods, preparing relevant pictures by designers is the most common one. However, it is always too subjective and the expressed image is always too superficial or limited, making it difficult to meet users’ real needs."
Explicitly reference the term of and problems associated with “Designer Fixation” and how it can differ from the actual user experience/requirements/needs.
"SmpleBorad Lab is an e-webpage platform that can be used for constructing image"
“SmpleBorad Lab” requires a reference: eg Establishing picture database for images board- An example for lifestyles of health and sustainability (LOHAS) image, Authors (in press)
The only reference I can find is to another article presumably you (based on the almost identical title and content) have had accepted for publication in Academia Journal of Scientific Research: https://www.academiapublishing.org/print/Liu%20et.%20al...pdf
Are these the same manuscripts and what is the connection?
It is acceptable to reference your own work but how is this latest submission that I am reviewing significantly different to the manuscript already accepted?
References
Observation only. Considering the novelty and recency of developments in this field as an area of research, only 5/25 references are from the last 10 years (2010-2020). It is good to see the work’s origins grounded in older, traditional, research areas, but are they more current topical references that this work should include – particularly in the field of participatory design for example?

Author Response
Dear Editor and Reviewers,
Many thanks for giving me an opportunity to revise our manuscript entitled “Establishing Picture Database for Images Board - An Example for LOHAS Image " (designs-830276). I appreciated the editor’s and reviewers’ positive and constructive comments and suggestions concerning my manuscript. These comments were all very helpful for revising and improving my paper, and also offered guidance in regard to the significance of our research.
The reviewers pointed out nine major problems:
Reviewer 2
- Explicitly reference the term of and problems associated with “Designer Fixation” and how it can differ from the actual user experience/requirements/needs.
- "SmpleBorad Lab's reference source, currently only the author's published papers, the published papers are different from the manuscripts currently submitted?
- Considering the novelty and recency of developments in this field as an area of research, only 5/25 references are from the last 10 years (2010-2020).
- 4 pages of English revision suggestions
Herewith please find the attached revised version. Modified text is displayed in red. The responses to the reviewers’ comments are appended below. If you have further comments, please let me know and we will do my best to make any necessary improvements.
Sincerely,
Peng-Jyun Liu
Institute of Applied Arts,National Chiao Tung University &
Department of Creative Product Design, Asia University
